# Deep Learning-based Heuristic Construction for Routing Problems with Dynamic Encoder and Dual-Channel Decoder Architecture

## Abstract

The routing problem is a classic combinatorial optimization challenge. Constructing heuristics using deep learning models presents a promising approach for its resolution. In this paper, we propose a novel model with a dynamic encoder and dual-channel decoder (DEDD) architecture to learn construction heuristics for the routing problem. The dynamic encoder encodes the node features of the decomposed subproblems at each selection step, thereby obtaining more accurate node embeddings. The dual-channel decoder facilitates more diverse node selections at each step, increasing the probability of the model identifying optimal solutions. Additionally, we design an effective node selection strategy to assist the model in choosing nodes at each step. Experimental results on the Traveling Salesman Problem (TSP) and the Capacitated Vehicle Routing Problem (CVRP) with up to 1000 nodes demonstrate that the solutions generated by the DEDD model are nearly optimal, underscoring its efficacy.

## 1 Introduction

The routing problem (Veres & Moussa, 2019) is a type of combinatorial optimization (CO) problem prevalent in logistics, distribution, transportation, and other fields with significant practical application value (Toth & Vigo, 2014). Its fundamental problems and classic variants include the Traveling Salesman Problem (TSP) and the Capacitated Vehicle Routing Problem (CVRP). The routing problem is NP-hard, making it particularly challenging to solve (Li et al., 2022). Current approaches for solving CO problems are mainly divided into four categories: exact approaches, approximation algorithms, heuristic algorithms and deep learning-based approaches. Exact approaches, such as branch-and-bound (Fischetti et al., 1994; Lawler & Wood, 1966) and dynamic programming (Bertsekas, 2000; Sniedovich, 2010), can theoretically find the global optimal solution but have exponential worst-case computational complexity. Thus, as the problem size increases, the computational effort required by these algorithms becomes impractical. Approximation algorithms (Williamson & Shmoys, 2011; Vazirani, 2001; Hochba, 1997) can obtain a theoretically guaranteed solution with polynomial computational complexity, but the solution quality is often suboptimal. Heuristic algorithms (Van Laarhoven et al., 1992; Wesley Barnes & LAGUNA, 1993; Halim & Ismail, 2019) involve experts designing specific heuristic rules based on extensive domain knowledge to search the solution space within an acceptable timeframe to find feasible solutions to CO problems. Due to the limitations of these three types of approaches, numerous deep learning models have been proposed in recent years to address some of the issues presented in the aforementioned approaches.

Learning-based neural combinatorial optimization approaches can be divided into two categories: learning improvement heuristics (Chen & Tian, 2019; Wu et al., 2021; Ma et al., 2021) and learning construction heuristics (Vinyals et al., 2015; Joshi et al., 2019; Kool et al., 2022). Learning improvement heuristics use deep reinforcement learning algorithms to learn the rules of iterative search operators, iteratively searching for solutions based on the learned rules. Essentially, learning improvement heuristics are iterative search algorithms with good optimization effects. The approaches proposed by Chen & Tian (2019) and Lu et al. (2019) have achieved results that match or even surpass professional combinatorial optimization solvers such as LKH3 (Helsgaun, 2017), Google OR tools

(Perron & Furnon, 2019), Gurobi, and Concorde (Applegate et al., 2006). However, these methods still lag behind end-to-end approaches.

Learning construction heuristics primarily use deep neural networks with an encoder-decoder structure. The encoder maps node information to embeddings, while the decoder, based on embeddings and other information, provides the probability of each node being selected at each step. Finally, based on these probabilities, a rule (e.g., greedily selecting the node with the highest probability Luo et al. (2023) determines the node chosen at each step. After providing the problem instance as input, the model repeatedly adds nodes to partial solutions until a complete solution is generated. Since this approach inputs the problem instance directly into the deep neural network and outputs the solution directly, it is also called an "end-to-end method." Compared to iterative heuristic algorithms, learning construction heuristics offer several advantages: First, they directly output the solution, resulting in faster solving speeds. Second, deep neural networks learn heuristic rules from data, replacing manually designed rules by experts. Finally, traditional CO problem-solving algorithms usually run on CPU, whereas deep neural networks run on GPU, enabling better parallelization and faster inference on large batches of problem instances. However, the quality of solutions directly output by deep neural networks often has room for improvement. Thus, various approaches (e.g., beam search (Nazari et al., 2018)) are employed to refine the initially generated solutions for better results.

To explore more ways to solve CO problems, scholars have proposed many deep learning models with different structures, trained through supervised learning (Fu et al., 2021; Joshi et al., 2022; Hottung et al., 2021a) or reinforcement learning (Hottung & Tierney, 2020; Hottung et al., 2021b; d O Costa et al., 2020). However, existing models still have some limitations. First, they are mostly confined to solving small-scale problems and perform poorly on larger-scale problems. This is because most models are trained and tested on specific scales (e.g., TSP100), leading to suboptimal generalization.

Second, existing learning-based neural combinatorial optimization models typically learn only one construction strategy, limiting the diversity of decisions when selecting nodes at each step. The decoder provides a probability matrix for node selection at each step, which is derived from the rules learned and the node embeddings output by the encoder. The model ultimately selects the node to visit based on the probability matrix and certain selection rules. In other words, the diversity of node selection strategies at each construction step mainly comes from the decoder's probability matrix and the node selection strategy based on it, which is insufficient for constructing high-quality solutions. Different node choices at each step lead to different directions for exploring complete solutions, so enriching the diversity of node selection strategies at each step is crucial.

Additionally, current deep learning models encode the problem instance into global node embeddings and other information embeddings once at the beginning. In subsequent construction steps, the decoder uses the same node embeddings to build the solution. However, when solving a routing problem with state transitions, using node embeddings from the previous state to solve the current state problem may lead to suboptimal strategies. Also, encoding all node at once may cause the model to learn scale-related features, performing well on trained scales but failing to capture necessary relationships among nodes in untrained scales.

To address these limitations, we propose a deep learning model with a dynamic encoder and dual-channel decoder (DEDD) structure for learning construction heuristics for routing problems. Firstly, to enhance model performance, the encoder in the DEDD model dynamically selects different nodes at each step to form new subproblems as inputs, allowing the decoder to choose the next node based on more accurate embeddings. Secondly, we propose a reasonable subproblem construction approach. At each construction step, we form subproblems from the starting point of the complete instance, the visited node from the previous step, and the remaining available nodes, balancing global and local information. Furthermore, since the dynamic encoder's input changes with construction steps, the DEDD model tends to learn scale-independent features, making it less sensitive to instance sizes and better at generalizing across different problem scales. Lastly, to enhance the effectiveness of node selection strategies, we propose a dual-channel decoder structure. The dual channels independently provide node probability matrices at each step, with the model selecting one channel's result to update the partial solution based on the probability matrix and selection strategy designed by us. Different channels share parameters in all but the final attention layer, thus enhancing strategy effectiveness while minimizing computational overhead.

It is worth noting that the DEDD model is not intended to completely surpass existing highly optimized professional routing problem solvers but to explore designing deep learning models that autonomously learn stronger heuristic rules for solving CO problems. We apply the model to solving TSP and CVRP of various scales, and experimental results demonstrate that the proposed DEDD model achieves good performance within shorter inference times and effectively solves instances up to a scale of 1000 nodes

## 2 RELATED WORKS

In 1985, Hopfield & Tank (1985) introduce the Hopfield network for solving the TSP and other CO problems, pioneering the use of neural networks for CO problem-solving. However, the Hopfield network requires retraining for each new TSP instance. In 2015, Vinyals et al. (2015) introduce the Pointer Network (PtrNet), the first to effectively employ deep neural networks for CO problems. The PtrNet features an encoder-decoder structure, with both components composed of long short-term memory (LSTM) networks. The model constructs complete solutions in an autoregressive manner through supervised learning. Given that labels are the optimal solutions of problem instances, and acquiring these solutions is costly, supervised learning becomes impractical. Bello et al. (2016) propose using the efficient reinforcement learning approach A3C to replace supervised learning, allowing the model to be trained on larger problem instances. Nazari et al. (2018) also use A3C to train and optimize the PtrNet. Kool et al. (2022) and Deudon et al. (2018) employ the Transformer architecture for CO problems. Kool et al. (2022) propose an attention model with an encoder-decoder structure. The encoder has three layers of attention, encoding all nodes simultaneously during training and inference. The decoder has a single layer of attention, utilizing a pointer-like attention mechanism for decoding. Since the proposal of the attention model (AM), numerous learning construction heuristics based on AM have been developed, but these methods can only perform well on small-scale CO problems.

In addition to deep learning models with encoder-decoder structures, some researchers have proposed approaches based on graph neural networks. Khalil et al. (2017) are the first to use the Q-Learning (Watkins & Dayan, 1992) algorithm of deep reinforcement learning to train graph neural networks for solving CO problems, achieving strong results in minimum vertex cover and maximum cut, but their results for TSP are less ideal. Ma et al. (2019) combine the PtrNet and graph neural networks by transforming graph neural network node features into node embeddings and using the PtrNet's attention mechanism to construct complete solutions, achieving strong optimization results in TSP. However, hierarchical reinforcement learning requires the setting of goals in order to achieve good performance, and the model has low efficiency in exploration during training.

Some researchers have used deep reinforcement learning to enhance search algorithms. Traditional search algorithms typically involve experts manually designing heuristic rules based on specialized domain knowledge. Currently, some researchers use deep reinforcement learning to enable models to automatically learn or select heuristic rules, iteratively searching for solutions based on learned

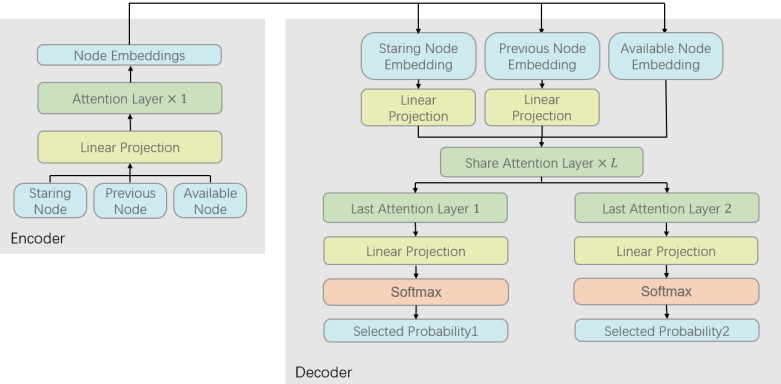

Figure 1: The architecture of the proposed DEDD model, featuring a single-layer encoder and a dual-channel decoder.

rules and obtaining higher-quality solutions after multiple iterations. Chen & Tian (2019) propose NeuRewriter to learn improvement heuristics, training two heuristic strategies to recursively refine initial solutions, achieving or surpassing the performance of professional solvers like LKH3 (Helsgaun, 2017). However, a limitation of this approach is the difficulty in parallelizing the solving of instances. Yolcu & Póczos (2019) adopt a local search framework, using deep reinforcement learning to learn variable selection operators, finding better-quality solutions in fewer iterations but with a long runtime. Additionally, many approaches that iteratively improve solutions still rely on expert-designed heuristic rules, and the iterative steps cannot be parallelized. Thus, learning improvement heuristics generally takes longer than learning construction heuristics. This paper focuses on learning construction heuristics for quickly solving instances.

## 3 MODEL

In this section, we propose a deep learning model with a dynamic encoder and dual-channel decoder structure and provide a detailed introduction to this model.

### 3.1 DYNAMIC ENCODER

In a VRP instance with $n$ city nodes, the node feature $x^i$ for node $i$, includes 2-dimensional coordinates and instance-specific features. For example, in TSP, the node coordinates are the node features. The model constructs a complete solution by sequentially selecting nodes. Using the same embeddings at each construction step may lead to poor performance. To address this issue, we use the starting point of the complete solution and the node selected in the previous step to bridge the partial solution with the subproblem composed of all available nodes, complementing the global information. These elements are input into the dynamic encoder along with the available nodes for re-embedding calculations at each step. Since the node from the previous step is input as the starting point of the subproblem, node selection in the current step also serves as a search direction problem for the optimal solution of the previous node. Additionally, incorporating the starting point and the previous node allows the model to dynamically learn the relationship between the partial solution and all available nodes at each step. As the scale of the subproblem input changes at each step, the model learns the relationships between nodes, making it less sensitive to the problem instance scale and thus achieving better generalization.

Recalculating node embeddings at each step increases computational cost. Most Transformer-based models have several times more encoder attention layers than decoder attention layers. With this structure, at each step, node embeddings must pass through a linear projection layer and multiple attention layers, leading to a significantly high computational cost. To address this, we propose a model with a DEDD structure, as shown in Figure 1. This model includes a dynamic encoder with one attention layer and a dual-channel decoder with multiple attention layers. This structure substantially reduces the computational overhead incurred by re-encoding node features in every construction step.

To express the node sequence $X_t$ at step $t$ (where $t \in \{1, 2, \ldots, n\}$ represents the current construction step), we denote the features of the node selected at step $j$ as $x_j$. Thus, $X_t = (x_1, x_{t-1}, x_t, x_{t+1}, \ldots, x_n)$. The dynamic encoder includes one linear projection layer and one attention layer. For an instance with $n$ nodes, the linear projection layer first transforms the node feature sequence $X_t$ into initial node embeddings $E^0 = (e_1^0, e_{t-1}^0, e_t^0, e_{t+1}^0, \ldots, e_n^0)$, where $e_j^0$ denotes the $d$-dimensional initial embedding of the node selected at step $j$. These initial embeddings are then processed through an attention layer to obtain the node embeddings

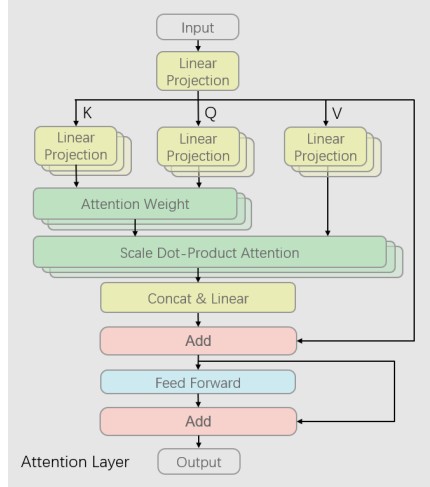

Figure 2: Multi-head Attention Layer

$E^1 = (e_1^1, e_{t-1}^1, e_t^1, e_{t+1}^1, \ldots, e_n^1)$. The attention layer consists of a multi-head attention (MHA) sublayer and a feedforward sublayer, as shown in Figure 2.

Let $E^{l-1} = (e_1^{l-1}, e_{t-1}^{l-1}, e_t^{l-1}, e_{t+1}^{l-1}, \ldots, e_n^{l-1})$ be the input of the $l$-th multi-head attention layer, so the MHA can be defined as follow:

$$Q_j^m, K_j^m, V_j^m = W_Q^m e_j^{l-1}, W_K^m e_j^{l-1}, W_V^m e_j^{l-1} \tag{1}$$

$$\begin{aligned} E^{l,m} &= \text{Attention}(Q^m, K^m, V^m) \\ &= \text{softmax}(Q^m K^{mT}/\sqrt{d_k})V^m, m = 1, 2, \ldots, M \end{aligned} \tag{2}$$

Where $E^{l,m}$ represents the node embeddings computed by the $m$-th attention head in the $l$-th multi-head attention layer. $Q$, $K$, $V$ are Query, Key, Value vectors, and $W_Q^m$, $W_K^m$, $W_V^m$ are linear transformation matrices of $Q$, $K$ and $V$, respectively. The $M$ attention heads respectively perform equations (1) and (2), with $d_k = d/M$.

After concatenating all $E^{l,m}$, they are multiplied by $W_O$ to obtain the output of the multi-head attention layer.

$$\text{Multihead}(Q, K, V) = \text{Concat}(E^{l,1}, E^{l,2}, \ldots, E^{l,M})W_O \tag{3}$$

$$\hat{e}_j^l = e_j^{l-1} + \text{Multihead}_j(Q, K, V) \tag{4}$$

$$e_j^l = \hat{e}_j^l + \text{FF}(\hat{e}_j^l) \tag{5}$$

Finally, using the skip connection layer (He et al., 2016) from equation (4) and the feedforward sublayer composed of two linear projection layers from equation (5), we obtain the output $e_j^l$ of the self-attention block. The process from equations (1) to (5) is represented as follows:

$$E^l = \text{MHA}(E^{l-1}) \tag{6}$$

## 3.2 DUAL-CHANNEL DECODER

To enhance the effectiveness of the node selection strategy at each construction step, we employ a dual-channel output decoder. The first $L$ attention layers of the dual-channel decoder are shared between both channels, while the final attention layer and linear projection layer in each channel have identical structures but do not share parameters. This design choice balances performance improvements with the increased computational cost. Utilizing one attention layer without shared parameters significantly enhances the model's performance with minimal additional computation.

As the final attention layer and linear projection layer in both channels do not share parameters, the same input may yield different outputs. The randomness from unshared parameters is the primary source of node diversity during the early stages of training, before the model is fully trained. After training, the diversity in node selection strategies primarily stems from the training data of the model. The trained model selects two optimal candidate nodes based on learned strategies at each construction step, rather than randomly choosing nodes of unknown quality, thus more efficiently finding higher-quality solutions. At step $t$, the encoder's output $E^1 = (e_1^1, e_{t-1}^1, e_t^1, e_{t+1}^1, \ldots, e_n^1)$ is fed into the decoder. The decoder then recalculates the embeddings for $e_1^1$ and $e_{t-1}^1$ using two linear projection layers, and concatenates them with $(e_t^1, e_{t+1}^1, \ldots, e_n^1)$ to obtain the input $\tilde{E}^0$ for its first attention layer. This process is expressed as follows:

$$\tilde{E}^0 = \text{Concat}(W_1 e_1^1, W_2 e_{t-1}^1, e_t^1, e_{t+1}^1, \ldots, e_n^1) \tag{7}$$

After passing through $L$ attention layers, we obtain the node embeddings $\tilde{E}^L$. Then $\tilde{E}^L$ passes through the last attention layer separately for each of the two channels:

$$\tilde{E}^1 = \text{MHA}(\tilde{E}^0) \tag{8}$$

$$\cdots\cdots$$

$$\tilde{E}^L = \text{MHA}(\tilde{E}^{L-1}) \tag{9}$$

$$\tilde{E}_1^{L+1} = \text{MHA1}(\tilde{E}^L) \tag{10}$$

$$\tilde{E}_2^{L+1} = \text{MHA2}(\tilde{E}^L) \tag{11}$$

$\tilde{E}_r^{L+1}$ is transformed by a linear projection layer into a vector $\mathbf{u}$, and finally undergoes a softmax transformation to obtain the selection probability $\mathbf{p}^t$, represented as follows:

$$u_i = \begin{cases} W_A \tilde{e}_j^{L+1}, & i \neq 1 \text{ or } 2 \\ -\infty, & \text{otherwise} \end{cases} \tag{12}$$

$$\mathbf{p}^t = \text{softmax}(\mathbf{u}) \tag{13}$$

The high memory and computational costs of the heavyweight decoder structure make it difficult to use reinforcement learning for training the DEDD model in this paper. Therefore, we use supervised learning to train the DEDD model. We define the loss function of the DEDD model as follows:

$$\text{loss} = -[\log(p_1^t) + \log(p_2^t)] \tag{14}$$

Where $p_r^t$ represents the probability predicted by channel $r$ of the DEDD model for the labeled node in step $t$.

### 3.3 CANDIDATE NODE SELECTION STRATEGY

The node selection process of the decoder is depicted in Figure 3. After obtaining the selection probabilities $\mathbf{p}^t$ from both channels, the node with the highest probability in $\mathbf{p}^t$ is selected as the candidate node for each channel (for instance, in Figure 3, the candidate nodes for instance 1 at step $t$ are 3 and 0). The merits of the two candidate nodes are compared, and the best one is chosen as the final node for step $t$. During the training process, data is fed into the DEDD model in batches. At each construction step, we calculate the loss for both channels across the entire batch, as shown in Equations (15) and (16), and select the nodes output by the channel with the lower loss as the final nodes for the entire batch. During inference, we calculate the additional distance for both channels across the entire batch, as indicated in Equations (17) and (18), and select the nodes output by the channel with the shorter distance as the final nodes for the entire batch. This approach evaluates the performance of both channels across multiple instances, thereby reducing the randomness associated with single-instance evaluation.

$$loss_1 = ls_{11} + ls_{12} + \cdots + ls_{1b} \tag{15}$$

$$loss_2 = ls_{21} + ls_{22} + \cdots + ls_{2b} \tag{16}$$

$$d_1 = d_{11} + d_{12} + \cdots + d_{1b} \tag{17}$$

$$d_2 = d_{21} + d_{22} + \cdots + d_{2b} \tag{18}$$

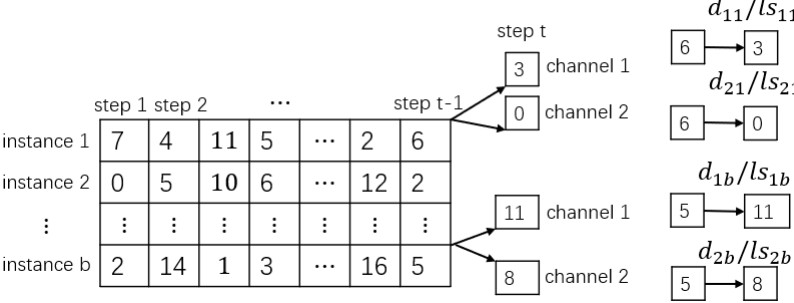

Figure 3: The node selection process for the $t$-th construction step.

### 3.4 DESTRUCTION-RECONSTRUCTION APPROACH BASED ON NODE SELECTION STRATEGY

To improve the quality of solutions constructed by the model at once, we adopt a destruction-reconstruction (DR) approach based on the node selection strategy. We perform operations on batches of instances. First, a segment, referred to as a partial solution, is randomly extracted from

the complete solution and input into the model for step-by-step reconstruction. In each reconstruction step, after the two channels output candidate nodes, the total distance values of all instances in the batch are calculated for both channels. The nodes output by the channel with the smaller total distance is selected as the final nodes for the current step. Once the partial solution is completely reconstructed, the distance of the partial solution before and after reconstruction is compared. If the reconstructed partial solution has a shorter distance, it replaces the original partial solution.

## 4 EXPERIMENT

In this section, we apply the proposed DEDD model to solve TSP and CVRP instances of various sizes. We compare its performance with other learning-based approaches and specialized solvers to assess its effectiveness. The TSP involves finding the shortest path for a salesman who starts from a city, visits all cities exactly once, and returns to the starting city, considering the distances between each pair of cities. The CVRP is a variant of the TSP where vehicles, starting from a depot and each with a capacity limit, aim to satisfy delivery demands for each city.

Our dataset is generated in accordance with the standard data generation procedure established in prior work AM (Kool et al., 2019). We employed the Concorde solver to obtain the optimal solutions for the TSP training set and utilized the HGS to acquire the optimal solutions for the CVRP training set. The training set includes one million TSP and CVRP instances, with sizes ranging from 4 to 100, respectively. The test set comprises 10,000 TSP and 10,000 CVRP instances of size 100, and 128 TSP and 128 CVRP instances of sizes 200, 500, and 1000.

Hyperparameter settings: We set the embedding dimension to 128, the encoder to 1 attention layer, the decoder to 5 parameter-shared attention layers, 1 non-shared attention layer. Each attention layer employs 8 heads for multi-head attention, and the feed forward layer dimension is set to 512. The DEDD model is trained separately on one million instances of size 100 for both the TSP and CVRP datasets. For training, we use a batch size of 1024 and the Adam optimizer with a learning rate of $10^{-4}$. For the TSP dataset, the learning rate decays by 0.97 per epoch, with training continuing for 150 epochs. For the CVRP dataset, the learning rate decays by 0.9 per epoch, with training lasting for 40 epochs. The DEDD model was trained and tested on an NVIDIA 3090 GPU, with the training to solve TSP instances requiring approximately 8 days and the training to solve CVRP instances requiring approximately 2 days.

We compare our approach to existing learning-based approaches and classical solvers. Classical solvers include Concorde (Applegate et al., 2006), LKH3 (Helsgaun, 2017), HGS (Vidal, 2022), and OR-Tools (Perron & Furnon, 2019). Learning-based approaches include LEHD (Luo et al., 2023), POMO (Kwon et al., 2020), MDAM (Xin et al., 2021), EAS (Hottung et al., 2021b), SGBS (Choo et al., 2022), BQ (Drakulic et al., 2023), and Att-GCN+MCTS (Fu et al., 2021). For optimality gap comparison, we use Concorde (Applegate et al., 2006) as the baseline for TSP and LKH3 (Helsgaun, 2017) as the baseline for CVRP. Our experiments focus on optimality gaps rather than inference time due to classical solvers running on CPU and learning-based approaches running on GPU, as well as potential differences in programming languages and platforms used for execution.

### 4.1 EXPERIMENTAL RESULTS

Table 1 presents experimental results comparing the DEDD model to various approaches. In TSP, the solutions directly generated by the DEDD model (greedy in Table 1) are of high quality and require minimal inference time. After 100 rounds of DR iteration, the DEDD model's performance surpasses all compared learning-based approaches in Table 1, except for LEHD (Luo et al., 2023). For TSP100, solutions with only 0.009% gap from the baseline are achieved after 100 rounds of DR iteration, nearly matching the baseline. For larger instances like TSP200, TSP500, and TSP1000, which are not in the training set, the DEDD model achieves gaps of approximately 0.03%, 0.2%, and 0.8% respectively, after just 300 rounds of DR iteration, closely approaching the baseline results. After 1000 rounds of DR iteration, the DEDD model nearly achieves optimal performance, with gaps of 0.001% for TSP100 and 0.015% for TSP200. The gaps for TSP500 and TSP1000 are 0.142% and 0.611%, respectively.

In CVRP, the DEDD model demonstrates superior performance relative to TSP. Due to the memory constraints of a single NVIDIA 3090 GPU, 10,000 CVRP100 instances are divided into two batches

Table 1: The experimental results for TSP and CVRP under uniformly distributed instances.

| Method | TSP100 | | TSP200 | | TSP500 | | TSP1000 | |
|---|---|---|---|---|---|---|---|---|
| | Gap | Time | Gap | Time | Gap | Time | Gap | Time |
| Concorde | 0.000% | 34m | 0.000% | 3m | 0.000% | 32m | 0.000% | 7.8h |
| LKH3 | 0.000% | 56m | 0.000% | 4m | 0.000% | 32m | 0.000% | 8.2h |
| OR-Tools | 2.368% | 11h | 3.618% | 17m | 4.682% | 50m | 4.885% | 10h |
| Att-GCN+MCTS | 0.037% | 15m | 0.884% | 2m | 2.536% | 6m | 3.223% | 13m |
| MDAM bs50 | 0.388% | 21m | 1.996% | 3m | 10.065% | 11m | 20.375% | 44m |
| POMO augx8 | 0.134% | 1m | 1.533% | 5s | 22.187% | 1m | 40.570% | 8m |
| SGBS | 0.060% | 40m | 0.562% | 4m | 11.550% | 54m | 26.035% | 7.4h |
| EAS | 0.057% | 6h | 0.496% | 28m | 17.080% | 7.8h | - | - |
| BQ greedy | 0.579% | 0.6m | 0.895% | 3s | 1.834% | 0.4m | 3.965% | 2.4m |
| BQ bs16 | 0.046% | 11m | 0.224% | 1m | 0.896% | 6m | 2.605% | 38m |
| LEHD RRC 1000 | 0.002% | 2.1h | 0.020% | 12.3m | 0.182% | 1.3h | 0.745% | 7.2h |
| DEDD greedy | 0.533% | 0.6m | 0.870% | 0.07m | 1.714% | 0.4m | 2.899% | 2.0m |
| DEDD DR 50 | 0.182% | 10.6m | 0.118% | 0.8m | 0.454% | 6.9m | 1.245% | 31.5m |
| DR 100 | 0.009% | 19.0m | 0.061% | 1.8m | 0.325% | 12.5m | 1.069% | 1.0h |
| DR 300 | 0.004% | 51.4m | 0.028% | 5.1m | 0.198% | 34.3m | 0.801% | 2.9h |
| DR 500 | 0.002% | 1.4h | 0.019% | 8.4m | 0.170% | 53.9m | 0.688% | 4.7h |
| DR 1000 | **0.001%** | 2.8h | **0.015%** | 17.0m | **0.142%** | 1.8h | **0.611%** | 9.4h |
| | CVRP100 | | CVRP200 | | CVRP500 | | CVRP1000 | |
| LKH3 | 0.000% | 12h | 0.000% | 2.1h | 0.000% | 5.5h | 0.000% | 7.1h |
| HGS | -0.533% | 4.5h | -1.126% | 1.4h | -1.794% | 4h | -2.162% | 5.3h |
| OR-Tools | 6.193% | 2h | 6.894% | 1h | 9.112% | 2.2h | 11.662% | 3h |
| MDAM bs50 | 2.211% | 25m | 4.304% | 3m | 10.498% | 12m | 27.814% | 47m |
| POMO augx8 | 0.689% | 1m | 4.866% | 7s | 19.901% | 1m | 128.89% | 10m |
| SGBS | 0.079% | 40m | 2.581% | 1m | 15.343% | 16m | 136.98% | 2.3h |
| EAS | **-0.234%** | 15h | 0.640% | 33m | 11.042% | 9.3h | - | - |
| BQ greedy | 2.993% | 0.7m | 3.527% | 4s | 5.121% | 0.4m | 9.812% | 2.4m |
| BQ bs16 | 0.611% | 10m | 1.141% | 0.6m | 2.991% | 6m | 7.784% | 39m |
| LEHD RRC 1000 | -0.100% | 2.7h | -0.346% | 14.5m | -0.01% | 1.35h | 2.484% | 7.8h |
| DEDD greedy | 3.557% | - | 3.055% | 0.1m | 2.877% | 0.4m | 6.456% | 2.2m |
| DEDD DR 50 | 0.525% | - | 0.402% | 1.4m | 0.948% | 6.0m | 4.149% | 38.7m |
| DR 100 | 0.257% | - | 0.123% | 2.5m | 0.606% | 10.4m | 3.602% | 1.1h |
| DR 300 | 0.007% | - | -0.196% | 6.8m | 0.136% | 32.8m | 2.688% | 3.0h |
| DR 500 | -0.071% | - | -0.302% | 11.5m | -0.015% | 57.8m | 2.332% | 4.9h |
| DR 1000 | -0.148% | 3.5h | **-0.446%** | 22.6m | **-0.245%** | 1.9h | **1.879%** | 10.0h |

of 5000 each for inference, and the results are averaged. Table 1 presents the total time after completing 1000 rounds of DR iteration for both batches. The solutions directly generated by DEDD show gap of about 3% from the baseline for CVRP100, CVRP200, and CVRP500, and about 6% for CVRP1000.

After 300 rounds of DR iteration, solution quality significantly improves. Specifically, the gap for CVRP100 is 0.007%, CVRP200's solution surpasses LKH3 (Helsgaun, 2017) by -0.196%, CVRP500's gap is close to the baseline at 0.136%, and CVRP1000 achieves a gap of 2.688%.

After 500 rounds of DR iteration, the DEDD model clearly outperforms LKH3 (Helsgaun, 2017) on CVRP100, CVRP200, and CVRP500. After 1000 rounds of DR iteration, the gap for CVRP1000 is also within 2%.

As shown in Table 1, the DEDD model's performance slightly fell short of EAS only in the CVRP100 solution. After 300 rounds of DR iterations, the DEDD model surpass five learning-based methods (except EAS and LEHD) and the traditional solver OR-Tools across four scales. For larger scales, such as CVRP200, CVRP500, and CVRP1000, the DEDD model demonstrate stronger capabilities, surpassing six learning-based methods (except LEHD) and OR-Tools with only 50 rounds of DR iterations. These results indicate that the DEDD model performs excellently.

## 5 ABLATION STUDY

We assess the effectiveness of various model components using TSP instances across four scales. Specifically, we compare the static encoder-dual channel decoder (SEDD) and the dynamic encoder-single channel decoder (DESD) with the dynamic encoder-dual channel decoder (DEDD) to evaluate the individual contributions of the dynamic encoder and dual-channel decoder to the model's performance. The dynamic encoder embeds nodes for subproblems at each construction step, while the static encoder embeds all nodes in a single initial step. The dual-channel decoder generates two probability matrices, from which higher-quality nodes are selected based on a node selection strategy, while the single-channel decoder outputs only one probability matrix. Table 2 shows that DEDD outperforms DESD and SEDD in all metrics for TSP100. However, for TSP200, TSP500, and TSP1000, the solutions output by DEDD are slightly inferior to those of DESD or SEDD.

First, the DEDD solutions may be slightly inferior to those of SEDD because models of two structures are trained on a dataset of one million TSP instances ranging from size 4 to 100, thus learning relationships among nodes in small-scale instances. When constructing complete solutions for TSP100, dynamic encoders improve performance comprehensively, as models of all three structures are trained on problems of related sizes. However, for larger-scale problems, neither dynamic nor static encoders learn to encode features of nodes at this scale during training, complicating the comparison of encoding approaches, as shown in the greedy metrics in Table 2.

Nonetheless, improvements can be made to the initial solution after it is directly output by the model. We continuously improve solution quality using the DR strategy. Since partial solution problems are present in the training set, the advantages and disadvantages of dynamic and static encoders are apparent. Specifically, a more suitable encoder can greatly improve partial solutions, thereby enhancing the overall solution quality. Table 2 shows that after multiple rounds of DR iteration, models using dynamic encoders (DEDD) outperform those using static encoders (SEDD) on all problem scales, proving the effectiveness of dynamic encoders.

Secondly, DEDD solutions may be slightly inferior to those of DESD for the following reasons. The dual-channel decoder offers richer node selection for each construction

Table 2: Experimental results of models with different architectures on uniformly distributed instances of TSP.

| | | TSP100 | | | | | |
| --- | --- | --- | --- | --- | --- | --- | --- |
| | | greedy | DR 50 | DR 100 | DR 300 | DR 500 | DR 1000 |
| DESD | gap | 0.554% | _0.023%_ | _0.013%_ | 0.0042% | 0.0024% | 0.0013% |
| SEDD | gap | _0.542%_ | 0.033% | 0.014% | _0.0041%_ | _0.0023%_ | _0.0013%_ |
| DEDD | gap | **0.533%** | **0.018%** | **0.009%** | **0.0040%** | **0.0022%** | **0.0012%** |
| | | TSP200 | | | | | |
| | | greedy | DR 50 | DR 100 | DR 300 | DR 500 | DR 1000 |
| DESD | gap | 0.922% | 0.121% | 0.075% | 0.033% | 0.023% | 0.0155% |
| SEDD | gap | **0.826%** | _0.119%_ | _0.068%_ | _0.029%_ | _0.021%_ | _0.0152%_ |
| DEDD | gap | _0.870%_ | **0.118%** | **0.061%** | **0.028%** | **0.019%** | **0.0151%** |
| | | TSP500 | | | | | |
| | | greedy | DR 50 | DR 100 | DR 300 | DR 500 | DR 1000 |
| DESD | gap | **1.684%** | 0.474% | **0.317%** | _0.210%_ | _0.188%_ | 0.165% |
| SEDD | gap | 1.732% | _0.455%_ | 0.329% | 0.224% | 0.190% | _0.163%_ |
| DEDD | gap | _1.714%_ | **0.454%** | _0.325%_ | **0.198%** | **0.170%** | **0.142%** |
| | | TSP1000 | | | | | |
| | | greedy | DR 50 | DR 100 | DR 300 | DR 500 | DR 1000 |
| DESD | gap | **2.713%** | **1.155%** | _1.081%_ | 0.828% | 0.693% | 0.616% |
| SEDD | gap | 3.091% | 1.381% | 1.181% | 0.944% | 0.837% | 0.726% |
| DEDD | gap | _2.899%_ | _1.245%_ | **1.069%** | **0.801%** | **0.688%** | **0.611%** |

step, but this does not guarantee a higher-quality complete solution. Thus, the results of a single solve may exhibit some randomness and yield slightly lower-quality solutions. However, after sufficient rounds of reconstructing partial solutions, randomness is greatly reduced, resulting in high-quality solutions. Table 2 shows that after 1000 rounds of DR, the quality of DEDD's solutions surpasses DESD, demonstrating the effectiveness of the dual-channel decoder structure.

Finally, we comprehensively compare the experimental results of the three structures. Since supervised learning-trained encoders may not accurately encode node features for problem scales outside the training set, and the solutions produced by dual-channel decoders in one solve may exhibit some randomness, we focus on the performance of models after multiple rounds of DR in the ablation experiment. Table 2 shows that after multiple rounds of DR, DEDD performs best among the three model structures, proving its effectiveness.

## 6 CONCLUSION

This paper introduces a novel model with dynamic encoders and a dual-channel decoder to learn construction heuristic for routing problems. The model uses supervised learning and incorporates dynamic encoders and a subproblem decomposition strategy tailored to routing problems, enhancing performance. The dual-channel decoder enables rich node selection and customized strategies, further enhancing performance. Experimental results show that the DEDD model performs exceptionally well in both TSP and CVRP, with particularly strong results in CVRP. Future work will focus on improving cooperative decision-making between decoder channels and exploring the potential of training the DEDD model with reinforcement learning.

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
