# OpenReview forum: "Deep Learning-based Heuristic Construction for Routing Problems with Dynamic Encoder and Dual-Channel Decoder Architecture"
_ICLR.cc/2025/Conference — Submitted to ICLR 2025_

### Official Review · Reviewer_rnAN · 2024-10-24

**Soundness:** 3
**Presentation:** 1
**Contribution:** 3
**Rating:** 3
**Confidence:** 4

**Summary:**

The paper addresses the challenging problem of combinatorial optimization in routing problems by proposing a deep learning-based heuristic approach. Specifically, it introduces an architecture called DEDD, which aims to learn effective heuristics for problems like the TSP and CVPR. The dynamic encoder enhances node embedding by re-encoding features at each construction step, while the dual-channel decoder increases diversity in node selection, improving solution quality. Experimental results show that the proposed model performs well for instances up to 1000 nodes, demonstrating nearly optimal solutions.

**Strengths:**

1. The design of the dynamic encoder is well-executed, enabling the model to produce more accurate node embeddings by re-encoding at each construction step. And the dual-channel decoder enhances node selection by maintaining two independent channels that share parameters in all but the final layer, leading to a richer and more diverse solution search space.

2. The experimental results are convincing, showing that DEDD can achieve near-optimal solutions for both TSP and CVRP, especially after the destruction-reconstruction  approach is applied.

**Weaknesses:**

1. The paper does not explicitly address how feasibility constraints are managed, particularly for the CVRP where vehicle capacities must be respected. The authors should discuss how they ensure that the solutions generated by DEDD are feasible with respect to capacity constraints.
2. The paper lacks a detailed discussion of the limitations of the proposed approach. It would be beneficial to provide a formal mathematical formulation that clearly defines the scope of problems that DEDD is designed to solve, including the assumptions and conditions under which the model operates effectively.
3. The authors highlight that a significant limitation of previous methods is their focus on small-scale problems and their poor generalization to larger or unseen instances. However, it remains unclear how the proposed DEDD model specifically addresses these issues. The paper lacks a detailed explanation of how the design choices within DEDD—such as the dynamic encoder and dual-channel decoder—are intended to overcome the problem of scaling and improve generalization.
4. The problem settings for CVRP instances are not fully elaborated. Important details such as the number of vehicles, vehicle capacities, and customer demands are missing. If these settings are based on a public dataset, it should be explicitly mentioned, and if they are generated, details on how the instances were constructed should be provided.
5. The overall writing quality of the paper is lacking, which significantly impacts readability and comprehension. Many crucial details are omitted, leaving gaps that force readers to make assumptions about the authors' intended meaning. Improving the writing, providing more detailed explanations, and avoiding ambiguity would greatly benefit the presentation.

**Questions:**

In addition to the points mentioned in the weaknesses section, I have some further questions and suggestions:

1. Could you clarify the training process for DEDD? Specifically, do you perform backpropagation after each construction step or only after generating the entire solution?
2. The term "labeled node" in Equation (14) is somewhat ambiguous. My assumption is that it refers to the node in the optimal solution corresponding to the current step. If this is correct, how do you handle situations where the labeled node is no longer available, such as when it has already been selected in previous steps?
3. In the reconstruction phase, it appears that a starting segment and an ending segment are given, and the middle part needs to be reconstructed. Unlike constructing the sequence from scratch, the ending segment may also contain valuable information. However, it seems that DEDD does not take the information from the ending segment into account during the reconstruction of the middle part.
4. The caption for figure 2 is missing.
5. Some key concepts in the paper are not adequately explained. For instance, the term "subproblem" is used without clear definition.

**Details Of Ethics Concerns:**

N.A.

---

> ### Author Response · Authors · 2024-11-27
>
> Thank you for your questions and feedback. Below, I will address the inquiries you have raised.
>
> 1. Regarding the handling of capacity constraints in the CVRP and the omitted details in this paper that you mentioned, we apologize for not providing a detailed explanation for every detail in the main text due to space limitations. Instead, we have chosen to highlight our main contributions within the confines of the limited space available. Concerning the handling of capacity constraints, we dynamically update the changes in the remaining capacity of vehicles to ensure the feasibility of the solutions. Our dataset was generated following the standard data generation procedure from previous work AM (Kool et al., 2019), using the Concorde solver to obtain the optimal solutions for the TSP training set and employing the HGS to acquire the optimal solutions for the CVRP training set. We have supplemented the relevant explanations in the updated PDF. Our contributions primarily lie in the proposal of a dynamic encoder and a dual-channel decoder; the dynamic encoder aims to achieve more effective node embeddings for problem-solving, thereby enhancing the model's performance. The dual-channel decoder is designed to enrich the candidate nodes at each construction step, thereby improving the model's performance. Thus, we mainly demonstrate the effectiveness of our proposed methods through ablation experiments, providing a clear and intuitive validation.
> 2. Regarding the training process of DEDD you mentioned, we perform backpropagation after each construction step. As for the label node in Equation (14), this is the label from the training set, which is the target the model learns. The entire training process can be summarized as follows: In each construction step, Channel 1 outputs the probability of each node being selected as a visiting node in the current construction step based on different node embeddings (including the starting point, the previous node, and the remaining unvisited nodes), forming a probability matrix. We mask the nodes that have already been visited, setting their corresponding probabilities to negative values. Then, we find the probability (p1) that the label node (e.g., Node 4) is predicted by the model to be the next visiting node in that construction step (i.e., the probability value corresponding to Node 4 in the probability matrix). Similarly, in Channel 2, we obtain the probability value (p2) for Node 4 in the second probability matrix. Ultimately, p1 and p2 form the loss function as described in Equation (14). That is, we aim for the probability of the label node being predicted as the next visiting node by the channels to be as close to 1 as possible in each construction step. Finally, during training, we use the label nodes to update the final visiting nodes in each construction step. That is, the complete solution constructed by the model during training is the complete solution from the training set.
> 3. Regarding your query about the reconstruction phase, I may not fully understand your meaning, so let me explain our approach again to see if it answers your questions. In the reconstruction phase, we extract a random-length partial solution from a complete solution at a random position, then input the partial solution into the model for reconstruction. If the reconstructed partial solution has better quality, we replace the original partial solution with the reconstructed one. Of course, this method of improving the partial solution treats the partial solution as a complete solution of an independent problem instance for improvement, which can effectively enhance the quality of the solution but may indeed overlook information in the complete solution that is valuable for improving the partial solution.
> 4. Regarding the definition of sub-problems, we have a clear explanation in line 186 of the paper.

---

### Official Review · Reviewer_AcEg · 2024-11-01

**Soundness:** 1
**Presentation:** 2
**Contribution:** 1
**Rating:** 3
**Confidence:** 3

**Summary:**

This paper proposes a supervised learning based method for finding the optimal path in the routing problem. The contribution of this work features a dynamic encoder and a dual-channel decoder, which are designed with the aim to increase diversity and get more updated information when making decisions. Results are evaluated in the Traveling Salesman Problem and the Capacitated Vehicle Routing Problem.

**Strengths:**

The idea of dynamic encoder and dual-channel decoder is interesting and novel. And extensive experiments were performed to compare the proposed method with the existing methods.

**Weaknesses:**

Major weaknesses:
1. The motivations behind the design choices and design details are not clear.
     - When designing the dynamic encoder, the starting point and the previous node are included to generate the new embedding. How much global information do the previous node and the starting point have? Would it be too weak to be helpful? This may be related to the ablation results presented in Table 2.
     - How is each channel of the decoder trained? Namely, how are p1 and p2 in Eq(14) decided? Also, the authors mentioned on Ln251 "After training, the diversity in node selection strategies primarily stems from the training data of the model." How is the diverversity ensured during the training?

2. How is the training/testing dataset collected? Is it a public dataset or generated by the authors? If generated by the authors, what's the protocol for generating these instances?

3. Statistical tests should be performed in Table 1 and Table 2 to show the significance of the comparison, as some of the results are very close.

4. It is mentioned in the paper that the computational and the memory consumption are very high. And it is the reason why reinforcement learning is not used. Could the author elaborate more why reinforcement learning means more memory and computations? Also, from Section 4, the encoder and decoder seem to be relatively small, with shallow layers and small embedding dimensions (128). It seems that the computational and memory costs are very affordable, even for RTX3090.

5. How does the performance change if the models are scaled up? For example, more channels in the decoder, or more layers?

Some minor issues:
1. Figure 2 does not have a caption.
2. On  Ln181, does the author mean "CVRP" instead of "VRP" ?

**Questions:**

See weaknesses.

---

> ### Author Response · Authors · 2024-11-27
>
> Thank you for your questions and feedback. Below, I will address the concerns you have raised.
>
> 1. Regarding the validity of the starting point and the previous node, we conducted separate ablation studies on these two points during our experiments. We found that if either of these nodes is removed individually or both are removed, the model's performance declines sharply in the initial stages of training. Even after complete training, the model's performance remains suboptimal. Therefore, we believe that information about the starting point and the previous node has a significant impact on our model's performance.
>
> 2. For the training of each channel in the decoder, the entire training process can be summarized as follows: In each construction step, Channel 1 outputs the probability of each node being selected as a visiting node in the current construction step based on different node embeddings (including the starting point, the previous node, and the remaining unvisited nodes), forming a probability matrix. We mask the nodes that have already been visited, setting their corresponding probabilities to negative values. Then, we identify the probability (p1) that the label node (e.g., Node 4) is predicted by the model to be the next visiting node in that construction step (i.e., the probability value corresponding to Node 4 in the probability matrix). Similarly, in Channel 2, we obtain the probability value (p2) for Node 4 in the second probability matrix. Ultimately, p1 and p2 form the loss function as described in Equation (14), and backpropagation is performed in each construction step. In other words, we aim for the probability of the label node being predicted as the next visiting node by the channels to be as close to 1 as possible in each construction step. Finally, during training, we use the label nodes to update the final visiting nodes in each construction step. That is, the complete solution constructed by the model during training is the complete solution from the training set.
>
> 3. Our dataset was generated in accordance with the standard data generation procedure established in prior work AM (Kool et al., 2019). We employed the Concorde solver to obtain the optimal solutions for the TSP training set and utilized the HGS to acquire the optimal solutions for the CVRP training set.We have supplemented the relevant explanations in the updated PDF.
>
> 4. Concerning your suggestion to use statistical tests to show the significance of comparisons, the results in Tables 1 and 2 are the average solutions of multiple problem instances in the test set.
>
> 5. You mentioned further explaining why reinforcement learning implies more memory and computation. Our analysis is as follows: Reinforcement learning consumes substantial device memory and computational resources when solving large-scale vehicle routing problems, primarily because these problems involve a large number of nodes and vehicles, leading to very high-dimensional state and action spaces. Complex deep learning models are required to handle this high-dimensional information, which themselves contain a large number of parameters that require substantial memory for storage. Moreover, to improve learning efficiency, reinforcement learning algorithms often need to store and process a large amount of historical data, further increasing memory demands. Additionally, to achieve effective learning, the algorithms require extensive computations, especially during the training phase, where they need to process large amounts of data and perform complex optimization processes, all of which place high demands on computational resources. Therefore, the demand for memory and computational resources naturally increases when reinforcement learning is applied to large-scale vehicle routing problems.
>
> 6. If the model uses more layers or more channels, it can indeed further enhance performance. The number of layers and channels currently employed in our model represents a trade-off between computational cost and performance. The performance gains achieved by increasing the number of layers or channels do not justify the increased computational costs.
>
> 7. Regarding the terminology issue in line 181 about VRP, by VRP we mean to collectively refer to problems such as TSP and CVRP. We aim to illustrate that the node features of VRP problems are composed of coordinates and specific features of different types of VRP. For instance, the node features for TSP are the nodes' two-dimensional coordinates, while the node features for CVRP consist of the nodes' two-dimensional coordinates and the nodes' demand quantities.
>
> 8. Concerning the principle of the decoder, as the explanation in the paper may not be intuitive, we have supplemented the updated PDF with schematic diagrams.

---

### Official Review · Reviewer_GpHi · 2024-11-04

**Soundness:** 2
**Presentation:** 3
**Contribution:** 2
**Rating:** 5
**Confidence:** 4

**Summary:**

This paper proposes a deep learning model, called DEDD, to solve combinatorial optimization problems such as traveling salesman problem (TSP) and capacitated vehicle routing problem (CRP). DEDD uses a dynamic encoder to update node embeddings at each step of the process and includes a dual-channel decoder for more diverse node selection strategies. Extensive experimental results for TSP and CVRP are presented to compare the performance of DEDD to standard baselines.

**Strengths:**

The paper was largely easy to follow and the experimental results included a suite of strong baselines.

**Weaknesses:**

1. The novelty/contribution of the method is weak given LEHD (Luo et al., 2023). The proposed method seems incremental compared to LEHD and DR (in the current paper) seems very similar to RRC (in LEHD).

1. A better motivation for the encoder and decoder structure would have helped to better understand the value of the proposed approach.

1. The rationale for the dual channel decoder is not explained clearly. Also, why only two channels and why not more?

1. In fact, in terms of performance, for both TSP and CVRP, LEHD is not that far off from the proposed DEDD. This further takes away from the value of DEDD.

1. In Table 1, for CVRP, HGS (Vidal, 2022) seems better than all solutions but this is not mentioned. Can you please comment on this?

1. Since HGS performs the best in the CVRP experiments, why not use HGS as the baseline reference, instead of LKH3?

1. Future work mentions exploring the training DEDD with RL. This seems like a straightforward approach to implement and I am curious why the paper did not do that. This might strengthen the contributions of the paper (see point 1 above).

1. Some practical considerations such as training time and computational complexity are not discussed in the paper. This would allow readers to better understand the paper's contributions and results in the context of existing work.

1. The paper needs to release code for all the presented results. Otherwise, how are we to know that the results in Table 1/Table 2 were computed correctly and fairly?

1. There are minor errors scattered throughout the paper. Please proofread the paper carefully. One such example is: (i) The caption of Figure 2 is not appropriate.

**Questions:**

See weaknesses above.

---

> ### Author Response · Authors · 2024-11-27
>
> Thank you for your questions and feedback. Below, I will address the concerns you have raised.
>
> 1. You mentioned the comparison between our method and LEHD (Luo et al., 2023), and suggested that we provide a better explanation for the motivation behind the structures of the encoder and decoder, to which I will now describe our thoughts. The work presented in this paper differs significantly from that of LEHD (Luo et al., 2023), specifically in terms of model architecture, generation strategies for node embeddings, solution construction strategies, and training and inference strategies, all of which are markedly distinct between LEHD and DEDD. Our primary contribution is the proposal of a dynamic encoder to achieve more effective node embeddings, thereby enhancing the quality of solutions. We also introduced a dual-channel decoder, allowing the model to have a wider selection of nodes at each construction step. Regarding your query on why we opted for only two channels, this decision represents a balance we struck between performance and the increase in computational complexity and parameter volume.
>
> 2. In response to your observation that the performance gap between LEHD and DEDD is not significantly large, it is indeed challenging to achieve substantial performance improvements given that the gap between LEHD and the baseline is already marginal. A direct comparison of the gap's numerical values may not highlight the performance enhancements of our model. However, when comparing the performance improvements of DEDD over LEHD in percentage terms, the value of our work becomes more apparent. The minimum performance improvement of our model is observed in the TSP200 instance, with nearly an 18% enhancement, while the maximum improvement is seen in the CVRP500 instance, with a  2350% increase in performance. Additionally, due to the volume of experimental data, a detailed comparison between DEDD and LEHD was not included in the paper. DEDD demonstrates a faster rate of solution quality improvement within fewer DR iterations compared to LEHD, which further underscores the performance of our model.
>
> 3. Regarding your comment that HGS outperforms all solutions listed in Table 1, this is indeed the case. However, HGS is a traditional method-based solver, whereas our approach is a learning-based constructive heuristic. Our focus is on comparing with other learning-based methods, hence the selection of LKH3 as the baseline does not affect the comparative results among learning-based methods.
>
> 4. As for your query on why we did not employ Reinforcement Learning (RL) to train our model, it is because RL-based methods may encounter critical issues such as sparse rewards and device memory limitations when tackling large-scale problems, making it highly challenging to train models directly on large-scale problem instances. Of course, training on small-scale problem instances would significantly reduce the difficulty.
>
> 5. The training time for the model has been supplemented in the updated PDF. The training time for TSP is approximately 8 days, and for CVRP, it is about 2 days.
>
> 6. The code has been submitted alongside the paper, and following the paper's acceptance, we will upload the code to GitHub.
>
> 7. The title of Figure 2 has been corrected.
>
> 8. Concerning the principle of the decoder, the explanation in the paper may not be intuitive. We have supplemented the updated PDF with a schematic diagram for clarity.

---

### Meta-Review · Area_Chair_Muqe · 2024-12-20

**Metareview:**

This paper addresses the classic combinatorial optimization challenge of routing problems by introducing a novel deep learning-based model called DEDD (Dynamic Encoder and Dual-Channel Decoder). Designed to learn construction heuristics for routing problems such as the Traveling Salesman Problem (TSP) and the Capacitated Vehicle Routing Problem (CVRP), DEDD introduces several innovative features to enhance solution quality and efficiency.

The dynamic encoder updates node embeddings at each step by encoding the features of decomposed sub-problems, providing more accurate and context-aware representations. Complementing this, the dual-channel decoder facilitates diverse node selection, increasing the likelihood of identifying near-optimal solutions. Additionally, the model employs a carefully designed node selection strategy to improve decision-making at each step.

Experimental results demonstrate DEDD’s effectiveness in solving large-scale TSP and CVRP instances. The model achieves near-optimal solutions for problems involving up to 1000 nodes. Compared to standard baseline methods, DEDD consistently outperforms in both accuracy and efficiency, highlighting its potential for real-world applications. By combining supervised learning with innovative architectural components, DEDD foresees advancing the development of deep learning-based heuristics for combinatorial optimization. Its ability to dynamically adapt embeddings and promote diversity in solutions positions it as a promising approach for solving complex routing problems in scalable and effective ways.

The three reviewers list a large number of weaknesses. I consolidate the lists and itemize the major ones:

1- Weak Novelty and Incremental Contribution: The proposed DEDD method appears incremental compared to prior work, such as LEHD (Luo et al., 2023), with DR in DEDD resembling RRC in LEHD. Additionally, the performance improvements over LEHD are marginal, which undermines the novelty and value of the contributions.

2- Lack of Motivation and Justification for Design Choices: The paper does not sufficiently motivate or justify key design decisions, such as the encoder and decoder structures, the rationale for using two channels in the decoder instead of more, or how diversity is ensured during training. Greater clarity and justification of these choices would strengthen the work.

3- Inadequate Baselines and Comparisons: The paper uses LKH3 as the baseline for CVRP, even though HGS performs better and is a more relevant reference point. The lack of statistical significance tests in Tables 1 and 2 further limits the rigor of performance comparisons.

4- Omission of Practical Considerations: Critical practical aspects, such as training time, computational complexity, and memory requirements, are not adequately discussed. These omissions prevent readers from fully understanding the feasibility of DEDD in comparison to existing methods.

5- Lack of Information about Dataset: The paper does not clearly explain how datasets were generated or whether public datasets were used. Furthermore, the absence of publicly released code raises concerns about the reproducibility and fairness of reported results.

**Additional Comments On Reviewer Discussion:**

The authors provided responses to the comments raised by the reviewers, which eventually did not lead to any change to the reviewers' ratings. I have gone over the responses and I also believe while some of the issues are clarified, the majority of the criticisms still hold, addressing which requires a major overhaul of the paper.

---

### Decision · Program_Chairs · 2025-01-22

Reject